# Emerging Evidence Supports Broader Definition of Chairside Behavior Guidance and Familial Compliance

**DOI:** 10.3390/healthcare12191935

**Published:** 2024-09-27

**Authors:** Paul S. Casamassimo

**Affiliations:** 1Division of Pediatric Dentistry, College of Dentistry, The Ohio State University, Columbus, OH 43214, USA; casamassimo.1@osu.edu; 2Nationwide Children’s Hospital, Columbus, OH 43205, USA

**Keywords:** behavior guidance, compliance, conceptual model

## Abstract

Background and Objectives: Behavior management as a set of clinical techniques to induce desirable clinical treatment and subsequent compliance behaviors in children and families varies internationally based on professional training, access to care, health literacy, and societal norms. This report proposes non-typical diagnostic considerations of additional inherent behavioral conditions and familial and social qualifiers that may help predict success both at the chairside and in compliance with home self-care behaviors to reduce caries susceptibility. Methods: A review of the medical and dental literature provides ample support for the consideration of changing characteristics of both the patient and the environment. Results: The current recommendations for choice of behavior guidance in direct clinical care, as used in the USA, often fall short of efficacy for a variety of reasons including the provider limitations, the extent and difficulty of treatment, and most recently appreciated, the complexity of negative childhood experiences, subclinical behavioral disorders, and immutable negative determinants of health outside the dental setting affecting interaction with health professionals. These same factors, such as family dysfunction and societal stresses, also impact compliance with out-of-clinic preventive efforts that many oral health care providers rely upon to help mitigate treatment limitations in reaching children. There are also behavioral elements of compliance and attitudes toward health that dentists need to recognize. Conclusions: A broader, more inclusive concept of behavior guidance to include factors beyond those typically associated with a dental patient affecting treatment and compliance with preventive behaviors may be beneficial. Every population and patient will have differing characteristics and require individualized care.

## 1. Introduction

Within about a year of one another, in the USA, two publications appeared in the scientific literature on dentistry. The first was issued by the National Institute of Dental and Craniofacial Research (NIDCR), entitled “Oral Health in America: Challenges and Opportunities” [1]. It was the culmination of several years of evidence-gathering on the contemporary state of oral health in the USA and was originally commissioned by the US Surgeon General. The other was a series of three evidence-based systematic reviews [2,3,4] commissioned by the American Academy of Pediatric Dentistry (AAPD) to determine the strength of evidence supporting chairside behavior guidance best practices [5] advanced by the AAPD and used throughout the world. While each approached the role of oral health behaviors from different directions, both ultimately challenged the traditional concepts of behavior guidance and set the stage for the consideration of a far more global view of how oral health professionals address the behavioral aspects of oral health care delivery.

The concept of compliance by families and caretakers with professional recommendations has long been simplified to “do as I say” without detailed consideration of mitigating social and environmental factors. Medicine in the USA and likely elsewhere has now recognized that health is not dictated by genetics alone and that the cumulated external influences—now termed the exposome [6]—have a major role in health and disease. Addressing those factors is critical to successful outcomes, whether it is eliminating them or working around them since many cannot be mitigated [7].

This paper proposes a more comprehensive view of chairside behavior guidance, as well as compliance with professional recommendations that engages a broader consideration of patient, family, cultural, societal, and health system factors when oral health professionals attempt to render care and create a patient-centered optimal oral health environment both when rendering care and post-treatment, in the interim between professional visits. Evolving evidence for a new approach will be presented to suggest why current techniques at the micro- and macro-levels have been largely unsuccessful. The author will use the theoretical framework of Fisher-Owens et al. [8], which they applied to dental caries initiation, but which can also be used to depict and explain the ever-widening orbits of influence affecting health behaviors. The term *behavior guidance* will be used to refer to chairside interaction between the child and the oral health professional, and *compliance* will be used to refer to post-treatment adherence to professional health maintenance recommendations.

## 2. A Model for Comprehensive Behavioral Management

The model proposed by Fisher-Owens et al. for caries initiation expands a singular view of the locus of causality to include orbits of influencing factors that may actually be the most powerful drivers of disease, either alone or in combination. In the case of dental caries, professional interventions too rarely depart from the oral cavity or adequately engage the family and community and their influences. When applied to behavior guidance, this model provides insight into the limits and failings of chairside motivation and cooperation and into long-term success in both disease control compliance and continuity of care. For example, traditional views of child behavioral wellness and family stability can prevent approaches to behavioral intervention from being successful. Simply illustrated, over two-thirds of adults have experienced a traumatic childhood experience [9] which may influence their interaction in a professional setting. Almost half have experienced more than one [10]. Similarly, forty percent of US children live in dysfunctional or challenged family environments [11]. In the first case, unknown or previously unmanifested anxiety or obstructive behavior may be resistant to traditional chairside behavioral guidance techniques. In the latter case, support for compliance with preventive recommendations may not exist within a child’s day-to-day living environment, making overall management of oral disease and its further prevention unlikely or impossible.

The model can be extended to the orbits of culture, community, and the health system. Oral health may not only not be a priority in some societal structures but may also be relegated to the management of pain only, without regard to the child’s psyche or long-term acceptance of preventive care. Emergency treatment of episodic dental pain becomes the norm. Additionally, as so aptly demonstrated by Hammersmith et al. [12], dental fatalism may be the societal oral health norm, so neither professional–patient compatibility nor home-based preventive behaviors have a chance in the population. Work by Burgette et al. [13] reinforces the depth of control of extended generations on oral health and its negative impact related to diet and perhaps dental fatalism. At the community level, access to care, as well as poor payment models for professional services, may hinder effective chairside behavior guidance, promotional preventive care, and case management services, all of which are aimed at structuring positive oral health behaviors at the patient and family levels. The reality of immutable factors at this level, such as community poverty and social disarray, is only now being appreciated in general health care [14]. Unfortunately, dentistry is decades behind medicine and seemingly unprepared to address this level of obstacles. The recent movement in industrialized countries to minimally invasive dentistry speaks to capitulation, related not only to effective chairside behavior guidance but also to seemingly unchangeable social and systems limitations [7].

## 3. Evidence for the Consideration of a New Paradigm

### 3.1. The Child

For decades, children seeking dental care were presumed to be all cut from the same cloth relative to developmental and emotional status unless they were accompanied by diagnoses. A variety of signs that children were not monolithic or manageable with simplistic Skinner-like behavior modification techniques began to emerge in this century. One notable example of this shift was the dramatic increase in definable autistic spectrum diagnoses with better screening and sub-typing [15]. The COVID-19 pandemic also laid open largely ignored or unidentified emotional dysfunction [16]. The pandemic’s toll on young psyches opened the medical community’s eyes to the far-ranging effects of social determinants of health on immediate and late-emerging behavior variants. Some of these behavioral variants have likely been in the child population for decades yet have been attributed to individual situational anxiety or other causes in the dental environment and have only recently been seen as permeating the wider child population and affecting dental care behavior. Historically, oral pain, extant or past, has been given credence in affecting behavior, and this too is often not given adequate credence as influencing the behavior of children confronted with a dental intervention [17].

### 3.2. The Family

When the emotionally healthy nuclear family was finally recognized as the exception rather than the rule is anyone’s guess. Single parenting, co-parenting by divorced couples, grandparenting, foster care, institutional parenting, and other variants now make up a larger proportion of childhood experiences. Even within the nuclear, two-parent family, the stresses of abuse, mental illness, incestuous relationships, and other pathological variants can exist and are often hidden. If we accept the premise that behavioral influences extend beyond chairside anxiety to include a child’s environment, then the challenges to effecting positive attitudes toward both the acceptance of dental care by children and aligning family beliefs toward oral health become more apparent. Recent work by researchers from the University of Pittsburgh about Appalachia on extended family influences on child oral health exemplifies the dynamics of family function [13].

### 3.3. The Community

The influence of community—also culture—on behavior at the micro- and macro-levels relative to oral health is seldom considered in manners other than a public health manner but is nonetheless important in individual, person-centered care. Provider gender, attire, physical contact, and expectations can all influence the acceptance of oral health care and a child’s reaction to treatment. Interestingly, this can be operative from both sides of the treatment equation. For example, historical biases suggest some cultures impart stoicism on even the very young, suggesting the likelihood of cooperation and freeing the provider from the responsibility to address fears. This has never been proved, and cultural mixing renders its veracity suspect. The work of Hammersmith et al., mentioned earlier, on dental fatalism also speaks to the cultural influence on preventive behaviors and compliance with desired oral health recommendations. Payne’s work in the educational arena [18] illustrates dietary modification challenges in compliance—the poor think of diet from the standpoint of survival, while the middle class views it from its health consequences, challenging health professionals to find the right balance for any family.

Even more fundamental is reduced or absent oral health literacy in some immigrant populations, making pain the portal for care-seeking and making even minimally invasive treatment impossible. Unless one is engaged in the care of diverse populations, another aspect of cultural influence—inexperience with technology and science resulting in limited understanding—is rarely considered. At the interpreter–provider–patient interface, for example, scientific explanation and cause-and-effect may have little or no meaning, hampering the acceptance of therapeutic inventions.

### 3.4. The System

The growth of dependence upon the Internet and alternatives to vetted scientific information and guidance influences chairside behavior and compliance, respectively. At the chairside, clinicians with little education in behavior guidance may meet a child whose parents have poorly prepared her for an interaction with an oral health care professional or who has been given misinformation about what to expect. Parents, influenced by similar misinformation in their own care and social milieu, may conflict with provider training and approaches. In the extension of care between visits, parents may seek care advice from peers or unknown sources, conflicting with advice and directives given by oral health professionals who are familiar with the family and those like them. Health disinformation is rampant [19]. The system may also limit access and resources. Direct professional patient care for general behavioral issues such as autism spectrum disorder or learning disabilities may be difficult to secure for families, but even more difficult may be support for behavioral management in the dental office, leaving few choices for a dentist except to refer.

## 4. What Is Next?

Table 1 offers a selective summary of some of the factors affecting both chairside behavior and compliance in a dental population in contemporary times, not always considered in the contemporary oral health space. These will, of course, vary by country, culture, and socioeconomic situation. More data are needed to clarify the extent and nature of these various orbits of influence. Medical care has already begun to assess these previously ignored factors. Primary care, in pediatrics, for example, mandates the assessment of dozens of conditions in children in the first few years of life [20]. Dentistry remains wedded to superficial health histories and is largely unlinked to comprehensive integrated health records that might shed light on important health concerns [21]. Medicine has similarly begun to integrate the assessment of the social determinants of health that influence health and compliance. A number of assessment tools are in use and hopefully can be used to generate data and ways to address obstacles in some populations. Larger medical systems have also begun to put into play adjunctive services like nutrition, social work, and community case workers to address immutable challenges to health issues like diabetes, obesity, and child abuse and neglect.

In dentistry, we are years away from the integration of the above realities into our care paradigm. More research is needed into the influence of these various orbits on chairside care and post-treatment compliance. It is unlikely that evidence from randomized clinical trials will emerge soon if at all. Associative data will likely drive change. Even then, how to successfully address these factors will need to be studied and validated. Optimum assessment will be challenging, and the application to management will vary from patient to patient and family to family.

It is beyond this paper’s aim to propose an algorithm for the management of the myriad of newly emerging factors influencing behaviors in oral health care at both the chairside and the post-treatment levels described here. Variations in the gross domestic product, educational levels achieved within the population, poverty, health literacy, culture, religion, and diet render the generalization of strategies across nations difficult if not impossible. This paper is an attempt to raise new issues that may explain chairside uncooperative behavior in children and the professional inability to manage it, as well as the failure of preventive post-treatment compliance by families. The wide variation across countries renders it difficult to offer more than a theoretical framework that can be applied anywhere. The model proposed by Fisher-Owens et al. [8] provides a framework for adaptation within nations to address both behavior guidance and compliance. The first step in addressing both challenges may be the recognition of the heterogeneity of the patients we care for and that one size does not fit all. With that realization, unfortunately, comes the challenges of determining how best to manage these individual situations, the need to modify our care and reimbursement systems, more engagement in meaningful social change through out-of-clinic advocacy, and better education of oral health professionals to function in a society that is heterogenous and changing.

The next step might be to engage stakeholders in an organized and directive process that identifies evidence in dentistry and supplements data with expert opinion, stakeholder input, reviews of the medical literature applicable to the topics, and investigation of electronic databases related to the topics. As has been largely the case in dentistry, evolution will likely be driven by trial and error and innovation, as randomized trials are limited and will likely require time and money. Dentistry would do well to follow medicine’s lead in addressing social determinants and system limitations to achieve health and equity. An analogous effort to this is the MORE Project [30], an integrative medical–dental model of care that presumes the benefit of physician–dentist interaction and has been implemented so that data can be derived and applied to sites other than those carefully structured in the project.

## 5. Conclusions

This paper posits a framework for future research and the structuring of chairside interventions, as well as improved compliance with the social determinants of health post-treatment. The complexity of and lack of evidence on factors influencing a child’s behavior during care and the subsequent adherence to professional recommendations, as well as international variance and the innumerable nuances affecting these activities, belie the proposal of a therapeutic model. These approaches will likely emerge, be tested in clinical research, and be specific to populations and their strengths and limitations and the resources available.

## Figures and Tables

**Table 1 healthcare-12-01935-t001:** Non-traditional Factors to Consider When Planning Chairside Behavior Intervention and Post-Visit Compliance.

Chairside Behavior Management	Post-Treatment Compliance
Child	
Adverse childhood experiences (ACEs) and neurodevelopmental problems [22]	Dental fatalism
Effects of deprived childhood environment	Peer interactions
Screen use; digital device preoccupation [23,24]	Food insecurity
Undiagnosed behavioral disorders	ACEs [25]
Suicidal thoughts [26]	Other medical costs due to ACEs [27]
Exposure to prenatal substance abuse [27]	
Exposure to environmental toxins in childhood	
Family	
Abusive environment	Deprived environment [28]
Dental fatalism	Low dental literacy
Authority figure ambiguity	Housing difficulty/homelessness
Culture/Community	
Bullying	Food desert; culture-driven diet
Health disinformation [19]	Dental fatalism
Lack of safety [28]	Lack of safety [28]
Poverty [29]	Poverty [29]
Health System	
Lack of behavioral services	
Lack of reimbursement for behavior intervention	

Note: This table is not intended to be all-inclusive but to provide directions for the reader to pursue additional information and evidence supporting the impact of various non-traditional factors on behavior and compliance as defined in this paper. As relationships emerge, they will need to be assessed for impact, as well as for ways to mitigate them.

## Data Availability

Not applicable.

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
