# Peer review of "Emerging Evidence Supports Broader Definition of Chairside Behavior Guidance and Familial Compliance"

_healthcare, 2024, doi:10.3390/healthcare12191935_

Round 1

Reviewer 1 Report

Comments and Suggestions for Authors

First of all, I would like to thank the authors for the effort they have put into researching this topic. However, I would like to recommend some considerations: 

- The article is focused on children or paediatric patients, I think you can value in the title to make reference to it or to the families. If not. You could consider another title as it does not develop a guide but simply presents a narrative article on the subject to be dealt with. 

- The introduction is reduced, understanding that it is a narrative article and it is usually longer than a clinical article. There are many variables to be taken into account, so the introduction needs to be expanded to better contextualise.

- The structure of the article is narrative, therefore, in my opinion, the last section should be longer and talk about other factors such as the type of country where the patients live, for example, if it is a developed country or not.

- The socio-economic level is fundamental for the development of a society. In developed countries it is taken into account that the health care system facilitates better access, even free of charge, to dental treatment. This means that children have a better state of oral health. It is very important to deal with this subject in your article and to expand and complete it with philosophical reflections due to the type of article that has been written.

- More tables like table 1 would be necessary to provide the reader with all the variables discussed in the texts, including socio-economic variables, race, age groups, country, etc. 

- I think there should be a conclusion section where all these characteristics and variables of human behaviour could be summarised.

Author Response

Dear Reviewer 1,

Comment: First of all, I would like to thank the authors for the effort they have put into researching this topic. However, I would like to recommend some considerations: 

Response: Thank you. I feel that is new ground for consideration in oral health care that we in dentistry are two decades behind on addressing!

Comment: - The article is focused on children or paediatric patients, I think you can value in the title to make reference to it or to the families. If not. You could consider another title as it does not develop a guide but simply presents a narrative article on the subject to be dealt with. 

Response: Author has tried to punctuate the report with reference to families and care-takers in both arms of the content – chairside behavior and post-operative compliance.

Comment: - The introduction is reduced, understanding that it is a narrative article and it is usually longer than a clinical article. There are many variables to be taken into account, so the introduction needs to be expanded to better contextualise.

Response: Author has tried to expand the introduction slightly and used available word count to address this and other reviewers’ comments throughout, hopefully accomplishing the same end as this recommendation.

Comment: - The structure of the article is narrative, therefore, in my opinion, the last section should be longer and talk about other factors such as the type of country where the patients live, for example, if it is a developed country or not.

Response: Author has added some commentary. The reviewer is absolutely right that there are variations based on income and other factors. To do justice to these differences would take a book! Author has chosen to address this comment by pointing out, as does the reviewer, the variation and that application of the proposed model to that specific population is the intent of the article.

Comment: - The socio-economic level is fundamental for the development of a society. In developed countries it is taken into account that the health care system facilitates better access, even free of charge, to dental treatment. This means that children have a better state of oral health. It is very important to deal with this subject in your article and to expand and complete it with philosophical reflections due to the type of article that has been written.

Response: Author recognizes these differences among countries. Even in developed countries, such as the author’s, there is a spectrum of oral health that can’t be dictated by a national average. In the US, for example, our lifestyle adopted by immigrants from poor countries paradoxically can negatively affect their previously good oral health! The gist of this report is the individualization of care that addresses challenges and uses strengths to their advantage.

Comment: - More tables like table 1 would be necessary to provide the reader with all the variables discussed in the texts, including socio-economic variables, race, age groups, country, etc. 

Response: Author recognizes the need to add material so has expanded the table.

Comment: - I think there should be a conclusion section where all these characteristics and variables of human behaviour could be summarised.

Response: Author has tried to do that generically for the sake of space and the reality that in the context of this report within an issue of the journal, presentation of a process is more important than content.

Author thanks the reviewer for these comments and hopefully the report is better for the responses!

Reviewer 2 Report

Comments and Suggestions for Authors

The subject matter addressed in the paper is innovative and offers fresh perspectives on the management of behaviour in children. In the introduction section, identify the shortcomings in the current literature and explain how the current work can address such deficiencies.

Although the title of the article referenced "Evidence Supports Broader Definition of Behaviour Guidance  and Compliance," the manuscript lacks the supporting evidence. It is recommended that the authors include a flow chart specifically related to the gathered evidence. Explain the compilation of the parameters presented in Table 1 and elaborate on their practical use in clinical practice.

Author Response

Dear Reviewer 2,

Comment: The subject matter addressed in the paper is innovative and offers fresh perspectives on the management of behaviour in children. In the introduction section, identify the shortcomings in the current literature and explain how the current work can address such deficiencies.

Response: Comments have been added to the introduction to address the above comment.

Comment: Although the title of the article referenced "Evidence Supports Broader Definition of Behaviour Guidance  and Compliance," the manuscript lacks the supporting evidence. It is recommended that the authors include a flow chart specifically related to the gathered evidence. Explain the compilation of the parameters presented in Table 1 and elaborate on their practical use in clinical practice.

Response: Author has modified title and abstract to indicate the emerging nature of data and clarifies that this is not a systematic review.

Reviewer 3 Report

Comments and Suggestions for Authors

It is a novel and necessary investigation. It is of interest to the readers. The topic presented is of great interest, and I acknowledge the effort and dedication invested in its development.

 However, I would like to offer some considerations that I believe could further enrich the content and strengthen the results presented. The title and abstract do not sufficiently express the study's subject,

the objective needs to be more focused

This paper proposes a more comprehensive view of chairside behavior guidance and compliance with professional recommendations that engages a broader consideration of patient, family, cultural, societal, and health system factors when oral health professionals attempt to render care and create a patient-centered optimal oral health environment both when rendering care, and post-treatment, in the interim between professional visits.

Interventions that should be made more explicit (behavioural and behaviour change theories across sources: consultations with experts from a multidisciplinary project advisory group, electronic databases, web searches, forward and backward searches of reference lists and hand searches of key behavioural journals.)

Author Response

Dear Reviewer 3,

Comment: It is a novel and necessary investigation. It is of interest to the readers. The topic presented is of great interest, and I acknowledge the effort and dedication invested in its development.

 However, I would like to offer some considerations that I believe could further enrich the content and strengthen the results presented. The title and abstract do not sufficiently express the study's subject, the objective needs to be more focused

Response: Author has restated in abstract and introduction a (hopefully) clearer objective.

Comment: This paper proposes a more comprehensive view of chairside behavior guidance and compliance with professional recommendations that engages a broader consideration of patient, family, cultural, societal, and health system factors when oral health professionals attempt to render care and create a patient-centered optimal oral health environment both when rendering care, and post-treatment, in the interim between professional visits.

Interventions that should be made more explicit (behavioural and behaviour change theories across sources: consultations with experts from a multidisciplinary project advisory group, electronic databases, web searches, forward and backward searches of reference lists and hand searches of key behavioural journals.)

Response: Great suggestion. Author has added mechanisms to come to solutions and clinical applications, and thanks to this reviewer’s suggestion, solved how to address another reviewer’s request to provide a rather extensive list of strategies to address these issues! Thanks!

Round 2

Reviewer 3 Report

Comments and Suggestions for Authors

The authors have adequately made the recommended changes